# Effect of relative income price on smoking initiation among adolescents in Ghana: evidence from pseudo-longitudinal data

Micheal Kofi Boachie ,[1] Mustapha Immurana,[2] Ernest Ngeh Tingum,[3] Noreen Dadirai Mdege,[4] Hana Ross [5]

For numbered affiliations see end of article.

**Correspondence to**
Dr Micheal Kofi Boachie;
mkboachie@gmail.com

## ABSTRACT

**Objective** Many smokers initiate smoking during adolescence. Making tobacco products less affordable is one of the best ways to control tobacco use. Studies on the effect of relative income price (RIP (ie, affordability)) of cigarettes on smoking initiation are scarce in low-income and middle-income countries, especially in Sub-Saharan Africa where data are limited. The goal of this study is to examine the effect of cigarette RIP on adolescent smoking initiation in Ghana.

**Setting** The study uses a pseudo-longitudinal data set constructed from the Global Youth Tobacco Surveys (GYTS (2000–2009 and 2017)) and RIP for the most sold cigarette brand in Ghana.

**Participants** The GYTS is a national survey on adolescents.

**Primary and secondary outcome** Effect of RIP on adolescent smoking initiation in Ghana.

**Results** Using the GYTS 2000–2009 data, we find that the probability of smoking initiation falls significantly in response to a higher RIP, with an elasticity of −0.372 (95% CI −0.701 to −0.042) for the unmatched sample and −0.490 (95% CI −0.818 to −0.161) for the matched sample. The RIP elasticity for women ((−0.888) (95% CI −1.384 to −0.392) and (−0.928) (95% CI −1.434 to −0.422)) is statistically significant at 1% in both the unmatched and the matched samples, respectively, while the RIP elasticity for men is statistically insignificant in the 2000–2009 surveys. Analysis of the 2017 GYTS shows a similar outcome: a negative relationship between RIP and smoking initiation, and the results are statistically significant for both men and women, and for both matched and unmatched samples.

**Conclusion** The affordability (RIP) of cigarettes is negatively related to the probability of smoking initiation among adolescents in Ghana. Raising tobacco taxes in line with income growth would make cigarettes less affordable and dissuade adolescents from initiating smoking.

## Strengths and limitations of this study

► This is the first study to explore the impact of Relative Income Price (ie, affordability index) of cigarettes on youth smoking initiation in Sub-Saharan Africa.
► Our analysis controls for variables that are known, empirically or theoretically, to be associated with smoking initiation, and the relationship is tested using a pseudo-longitudinal data set of 17 years.
► We also examine potential sex differences in the effect of affordability on cigarette smoking initiation: this is key to the implementation of tobacco control policies that confer adequate protection across both genders.
► Since Global Youth Tobacco Surveys data are available in many low and middle-income countries (LMICs), our study provide a template on how to do analyses elsewhere in order to enhance our understating of the impact of cigarette affordability on smoking uptake in LMICs.
► The results are subjected to self-reporting and recall biases as well as omitted variable bias due to lack of data on other factors affecting smoking uptake.

huge financial burdens on households and governments.[4 5] The majority (80%–90%) of the one billion adult smokers globally began the habit during their adolescence.[6] The current smoking prevalence in Ghana is still low relative to other African countries (3.2% among adults aged 15 years and older,[7] and 6.4% among students aged 13–15 years).[8] However, the number of smokers is predicted to increase from 1.3 million to 1.7 million (ie, by about 30%) between 2020 and 2025.[9] The expected increase in the number of smokers will be partly driven by initiation among adolescents, therefore lowering initiation is key to slowing down the tobacco epidemic in Ghana.

Empirical evidence shows that tobacco consumption (initiation and intensity) is significantly inversely related to price.[10–12]

## INTRODUCTION

Tobacco use is one of the major risk factors for many non-communicable diseases such as lung cancer and ischaemic heart disease,[1 2] and it accounts for over eight million deaths annually worldwide.[3] Tobacco use also imposes

In addition to being the most cost-effective measure to reduce tobacco use, increasing the taxes on tobacco products generates revenue for governments.[13–15] Nevertheless, economic factors such as income growth can negatively affect the response of tobacco consumption to tax/price changes.[16 17] Increasing tobacco prices can be more effective in reducing tobacco consumption if it reduces affordability.[18] Affordability (relative income price (RIP)) elasticity, which measures the sensitivity of consumers to real changes in both price and income, can, therefore, be a useful parameter to explain and predict the sensitivity of consumers to tobacco tax and price policies even in the presence of income growth.[17] This is particularly important for tobacco control measures aimed at adolescents because they are more price sensitive than adults,[12 19] for instance in Ghana where 71.3% of current cigarette smokers aged 13–15 buy their own cigarettes.[8]

Some Sub-Saharan African (SSA) studies, although few, have demonstrated that increasing cigarette prices reduces smoking prevalence and intensity of use.[12 20–25] However, there is a scarcity of studies on the relationship between cigarette prices and smoking initiation in the SSA context. One study, using data from 48 countries, including four from SSA (Kenya, Nigeria, Senegal and South Africa), concluded that higher cigarette prices reduce smoking initiation in early youth, with girls being more responsive than boys.[26] However, findings from the SSA countries were not reported separately from the overall study findings. A study in South Africa reported a significant reduction in regular smoking initiation among men due to higher cigarette prices, but not among women.[10] Another study in Nigeria and Ghana concluded that increasing cigarette prices resulted in a reduction in both 30-day cigarette smoking and cigarette smoking onset in both countries.[27]

Ghana has implemented a number of tobacco tax changes over the last 20 years. For example, it implemented a specific excise tax in 2008 and subsequently switched to an ad valorem tax structure in 2010.[28 29] At the same time, per capita income in Ghana has been growing at an average rate of 4.4% annually in the last decade.[30] These changes have implications for the retail prices of tobacco products (eg, cigarettes) and the affordability of cigarettes or other tobacco products. However, to our knowledge, no study has analysed the impact of cigarette affordability on smoking initiation in Ghana.

We address this critical evidence gap by examining the effect of cigarette affordability on smoking initiation among adolescents in Ghana. We hypothesise that making cigarettes less affordable reduces the likelihood of smoking initiation among young people, and we make use of the Global Youth Tobacco Survey (GYTS) data and other data sets to test that hypothesis. The control variables used are sex, age, parents' and friends' smoking status, being offered a cigarette for free, family/class discussion about tobacco, exposure to antismoking messages and exposure to tobacco advertisements. These controls are based on variables that are known, empirically or theoretically, to be associated with smoking initiation.[27 31]

Our analysis addresses the potential endogeneity of price, if any, as a driver of cigarette demand through (1) using aggregate level prices and not self-reported prices[32] and (2) the use of propensity score matching (PSM) techniques.[33 34] We also examine potential sex differences in the effect of affordability on cigarette smoking initiation. An understanding of these dynamics is key to the implementation of context-specific tobacco and noncommunicable disease control policies in Ghana.

## MATERIALS AND METHODS
### Data and variables
We make use of three waves (2000, 2006 and 2009) of the GYTS and RIP (affordability) data (1991–2009) to analyse the effect of affordability on smoking initiation among adolescents in Ghana. The WHO defines adolescents as young people between the ages of 10 and 19 years. The GYTS questionnaire specifies ages from 11 years or younger to 17 years or higher for current age (ie, age at the time of survey). It also captures age at first puff, which ranges from 7 years or younger to 16 years or older. For the purposes of this study, we classify respondents as adolescents, youth or young people. The terms are used interchangeably in the study. The GYTS is a school-based survey developed to enhance the capacity of countries to monitor tobacco use among the youth as well as implement and evaluate tobacco control and prevention programmes.[35] The GYTS is a cross-sectional survey and does not follow individuals over time, but provides data on smoking patterns among adolescents. In countries where it is conducted at regular intervals, it allows the monitoring of trends over time. We are aware of the 2017 GYTS for Ghana, but we do not include it in the analysis of the pooled 2000–2009 surveys due to inconsistencies in the questions asked, compared with those in previous GYTS surveys. There is no other survey on adolescents in Ghana with comparable measures except the ones outlined. We analyse the 2017 GYTS separately while linking it with RIP data for 2008–2017 based on age-at-risk criteria[27 32 36] (as done for the 2000–2009 surveys).

Although the GYTS data contain adolescents whose first puff was at age 7 or younger, we assume that a student is at risk at age 8 because that is the age at which the child is relatively developed and is able to start out-of-home interaction with peers.[27 32 36] Students who started smoking before reaching age 8 and those below age 8 are, therefore, excluded from the pooled sample and not followed. In line with previous studies, a student exits the sample once smoking is initiated.[10 27 32 33]

In Ghana, there was no law prohibiting the sale of cigarettes to minors until 2012 when restrictions on sale to persons below age 18 years were introduced.[37] The GYTS sample is drawn using a two-stage cluster-sampling design.[35 38 39] Schools are selected with probability proportional to school enrolment size during the first stage, and

then classes within participating schools are selected as a systematic equal probability sample with a random start during the second stage. All students in the selected classes are eligible to participate in the survey.

The Ghana GYTS questionnaire captures information on the use of tobacco products such as cigarettes and shisha. The data also include information on parental and peer smoking habits, perception about tobacco use (eg, weight gain, health effects and ease of quitting), money spent on tobacco in the last 30 days before the survey and secondhand smoking (SHS).[40] Studies vary widely on the way they define or measure smoking initiation.[41–43] For the GYTS, smoking initiation is measured using the definition of a lifetime smoker, that is, a person who has ever tried smoking, *even one or two puffs of a cigarette*.[27 43 44] Thus, for our study, smoking initiation (dependent variable) is a dichotomous variable generated from the following GYTS question where students answer Yes/No: *Have you ever tried or experimented with cigarette smoking, even one or two puffs?*.[27]

The main independent variable is the affordability index or the RIP, measured as the percentage of GDP per capita (per capita income) required to buy 100 packs of cigarettes (20 sticks per pack, in total 2000 sticks).[17 18 45 46] Affordability is a relative measure and is calculated using nominal prices and nominal GDP per capita, or real prices and real GDP per capita. Data on per capita income are obtained from the World Bank's World Development Indicators,[30] and those of average cigarette prices (for the most-sold brand) come from the WHO, relevant publications of the tobacco industry[47] and the Government of Ghana.[48] Years with missing data on prices were interpolated using the formula:

$$P_{t-1} = \frac{P_t}{(1+Tob.Inflation_t)} \qquad (1)$$

where $P_{t-1}$ is the previous year's price of cigarette, $P_t$ is current price of cigarette and $Tob.Inflation_t$ is the current tobacco inflation.[49] We then calculate RIP following methods used by preceding studies, with a lower affordability index (RIP) indicating that cigarettes have become more affordable and a higher value indicating that cigarettes have become less affordable relative to the reference year.[17 18 45 46] The WHO uses the same approach to obtain its affordability index. Other independent variables used are sex, age, parents', and friends' smoking status, whether offered cigarettes for free, family/class discussion about tobacco, exposure to antismoking messages, and exposure to tobacco advertisements. These variables are selected as they have been shown, theoretically or empirically, to be associated with smoking initiation.[27 31] Except age and RIP, which are continuously measured, all variables are dichotomous.

## Data analysis

We construct a pseudo-longitudinal dataset based on current age and age at first puff. In doing this, we create a historical dataset starting from age 8 (age-at-risk criteria) and follow the person until s/he initiates smoking. This is done by inferring the year of first puff using the GYTS question: "How old were you when you first tried a cigarette?" and the age at the time of the survey.[27]

STATA routine command, *expand*, is used to expand the person's age at the time of the survey after which an event variable indicating smoking status is created. We then link the RIP (affordability index) data to this pseudo-longitudinal dataset.

Our statistical technique is duration or event history analysis where the timing of transition into initiation is a function of the probability of initiating in period *t* conditional on not having experienced a transition until period *t*, known as the hazard rate.[10] Following previous empirical studies,[10 32 36] we employ the discrete time-hazard model, with logit specification (see equation 2), to study the association between RIP (affordability) and smoking initiation among adolescents.

$$Pr\left(Initiation = 1|RIP, X^{'}\right) = \beta_0 + \beta_1 RIP + \beta_i X^{'} \qquad (2)$$

where Initiation is defined as first cigarette puff, RIP is the affordability index, $X^{'}$ is a vector of other independent variables affecting smoking initiation among adolescents and $\beta$ is a vector of the regression coefficients. The predictors, $X^{'}$, represent age, sex, whether offered free cigarette, parental and peer smoking status, family/class discussion on the dangers of tobacco, exposure to tobacco advertisements and hearing of antismoking messages and awareness of tobacco control policies introduced in 2012 (for the 2017 GYTS). We report ORs, and the statistical level of significance is set at p≤0.1. OR <1 implies that when a higher share of income is required to buy 2000 cigarettes (cigarettes are less affordable), the risk of smoking initiation declines, and vice versa. The partial derivative of equation 2 with respect to RIP gives the affordability elasticity.

To check the robustness of the logistic regression estimates, we employ a PSM technique to match ever-smokers to never-smokers based on the propensity scores. Our approach to matching follows previous studies.[33 50 51] The propensity scores are obtained by running a logit regression to estimate the probability of being a smoker based on the variables in equation 2, except RIP, and the predicted probabilities are then used to match ever-smokers to never-smokers. Using the neighbourhood matching, ever-smokers are matched to their two nearest neighbours. After matching the sample, we re-estimate the logit model to assess the effect of affordability on the probability of initiating smoking, using GYTS weights on the matched sample.[50 51] With the matching approach, we are able to obtain the effect of affordability on the probability of initiating smoking among adolescent smokers and non-smokers who possess similar characteristics based on the propensity scores. This technique addresses issues of endogeneity and concerns relating to the fact that some never-smokers will never choose to smoke or use any form of tobacco irrespective of market conditions.[33 34] Furthermore, we minimise the problem of endogeneity by not

using self-reported prices.[32] Data analysis is conducted using STATA V.15. The study benefited immensely from discrete time modelling guidelines and STATA code produced by Professor Stephen Jenkins.[52]

## Interpretation of RIP elasticity

Although the RIP is measured in percentages, interpretation of the affordability index follows the same procedure for elasticity interpretation. The elasticity measures the percentage change in probability of initiating smoking following a percentage change in RIP, ceteris paribus. Assuming the current RIP is 6% or 0.06, then a 1% increase in RIP corresponds to the current RIP increasing from 6% to 6.06%. When using a unit change interpretation, a unit change will be RIP moving from 6% to 7% and, therefore, probabilities will change in absolute units and not percentages. Such distinction is important in understanding the impact of affordability on smoking behaviour.

## Patient and public involvement

No patient involved.

## RESULTS
### Descriptive statistics

A total of 20 202 adolescents were interviewed across the three GYTS waves (2000–2009). Fifty-four per cent of the respondents were men, while 76.47% were aged 15 years or less. In the 2017 GYTS, 5664 people were interviewed, with about 48% being men. Overall, 12.46% and 8.9% of the respondents in the pooled (2000–2009) and 2017 surveys, respectively, had ever smoked.

Given our age-at-risk criteria, 15 861 (2000–2009 GYTS) and 5389 (2017 GYTS) people were eligible for inclusion in our pseudo-longitudinal analysis. For surveys prior to 2017, 4.2% initiated smoking at some point between 1991 and 2009, and about 77% of smoking initiators did so before reaching age 16. Furthermore, 67% of initiators were men. Overall, men represented 53.62% of the eligible respondents. In the 2017 survey, 4.72% of the respondents initiated smoking at some point between 2008 and 2017. The characteristics of the samples are presented in table 1. Due to incomplete information on some of the variables, the number of people used in the regression varies.

### Regression results

Results from the logit regressions for the unmatched and matched samples are reported in table 2 (GYTS 2000 – 2009) and 3 (GYTS 2017). The results show a statistically significant and negative relationship between RIP and smoking initiation. For instance, the results for the full unmatched sample (table 2) show that OR on RIP is 0.98. This implies that a unit increase in affordability is associated with 0.98 odds of initiating smoking (OR=0.98, p<0.05). Thus, an adolescent who is subjected to a unit increase in RIP has 2% (ie, 1–0.98)×100 = 2%)) lower

| Table 1 | Descriptive statistics | |
|---|---|---|
| **Variable** | **2000–2009 GYTS** | **2017 GYTS** |
| | **Students, n=15 861** | **n=5389** |
| Initiated smoking during the period | 4.20% | 4.72% |
| RIP (affordability) | 19.87 (SD=6.53) | 7.63 (SD=0.86) |
| Offered free cigarettes | 12.44% | 8.13% |
| Sex (male=1) | 53.62% | 48.73% |
| At least one parent smoke | 11.78% | – |
| Family/class discuss about tobacco | 72.50% | 51.47% |
| At least a friend smoke | 15.94% | – |
| Exposed to tobacco adverts | 40.46% | 56.03% |
| Age (years) | 14.15 (SD=1.7) | 14.10 (SD=1.03) |
| Heard anti-smoking campaigns | 74.64% | 57.26% |
| Age at initiation (years) | 11.95 (SD=2.9) | 11.26 (SD=2.41) |
| Percentage of initiators before age 16 | 77% | 94% |
| Percentage of initiators who are males | 67% | 59% |
| Awareness of smoke free policies | – | 78.24% |

GYTS, Global Youth Tobacco Surveys; RIP, relative income price.

probability of initiating smoking than his/her counterpart who is not subjected to the same increase. Note that these results are not elasticities. Women have 40.1% (ie, 1–0.599)×100 = 40.1%)) lower probability of initiating smoking (OR=0.599, p<0.01) compared with their male counterparts in the unmatched sample.

Similarly, in table 3, in the unmatched sample, an adolescent faced with a unit increase in RIP has about 18% (ie, 1 – 0.821) × 100 = 17.9%)) lower probability of starting smoking than his/her counterpart who is not subjected to the same increase.

Other significant factors that influence smoking initiation in our two samples include whether the adolescent's parents (OR=2.131, p<0.01) or friends (OR=4.109, p<0.01) smoke. In addition, adolescents who are offered free cigarettes have a high probability of initiating smoking (OR=1.491, p<0.01) compared with those who receive no such offer for the 2000–2009 wave (table 2). In the 2017 wave, the odds of adolescents starting smoking when given cigarettes freely is 3.403 (p<0.01) (table 3).

In the matched sample (table 2), 611 ever-smokers were matched to their two nearest neighbours (1000

**Table 2** Effect of RIP on smoking initiation among adolescents (GYTS 2000–2009)

| Variables | Unmatched OR | Matched OR |
|---|---|---|
| Affordability (RIP) | 0.981† | 0.974* |
| | (0.009) | (0.009) |
| Offered free cigarette (ref=no) | 1.491* | 0.517* |
| | (0.216) | (0.071) |
| Sex (ref=male) | 0.599* | 0.615* |
| | (0.072) | (0.076) |
| At least one parent smokes (ref=no) | 2.131* | 0.862 |
| | (0.280) | (0.104) |
| Family/class discussion (ref=no) | 1.001 | 1.711* |
| | (0.133) | (0.230) |
| At least one friend smokes (ref=no) | 4.109* | 1.094 |
| | (0.520) | (0.126) |
| Exposure to adverts (ref=no) | 1.155 | 1.027 |
| | (0.140) | (0.121) |
| Age | 1.150* | 0.991 |
| | (0.042) | (0.031) |
| Heard of anti-smoking message/campaign (ref=no) | 1.342‡ | 2.048* |
| | (0.217) | (0.321) |
| Survey cycle (ref=2000) | | |
| 2006 | 0.958 | 0.880 |
| | (0.146) | (0.138) |
| 2009 | 1.108 | 1.003 |
| | (0.171) | (0.159) |
| Log (time) | 1.110 | 1.393* |
| | (0.106) | (0.146) |
| Constant | 0.000* | 0.048* |
| | (0.000) | (0.024) |
| Observations | 106 673 | 10 078 |
| Number of people | 15 201 | 1611 |
| Ever-smokers | 611 | 611 |
| Pseudo $R^2$ | 0.0815 | 0.0448 |
| $\chi^2$ | 439.2* | 91.84* |

Robust standard errors in parentheses.
*p<0.01.
†p<0.05.
‡p<0.1.
GYTS, Global Youth Tobacco Surveys; RIP, relative income price.

**Table 3** Effect of RIP on smoking initiation among adolescents (GYTS 2017)

| Variables | Unmatched OR | Matched OR |
|---|---|---|
| Affordability (RIP) | 0.821† | 0.804* |
| | (0.066) | (0.065) |
| Sex (ref=male) | 0.659† | 0.902 |
| | (0.120) | (0.177) |
| Offered free cigarettes (ref=no) | 3.403* | 0.978 |
| | (0.726) | (0.221) |
| Heard of anti-smoking message (ref=no) | 1.165 | 1.009 |
| | (0.213) | (0.192) |
| Exposed to tobacco adverts (ref=no) | 3.030* | 1.893* |
| | (0.622) | (0.421) |
| Smoke free policies awareness (ref=no) | 1.250 | 1.160 |
| | (0.329) | (0.332) |
| Age | 1.847* | 1.793* |
| | (0.294) | (0.279) |
| Class discussion on tobacco harms (ref=no) | 0.795 | 1.278 |
| | (0.138) | (0.241) |
| Log(time) | 0.104* | 0.119* |
| | (0.067) | (0.076) |
| Constant | 0.000* | 0.002* |
| | (0.000) | (0.002) |
| Observations | 37 654 | 4850 |
| Number of people | 5301 | 747 |
| Ever-smokers | 231 | 206 |
| Pseudo $R^2$ | 0.0599 | 0.0292 |
| $\chi^2$ | 158* | 38.72* |

Robust standard errors in parentheses.
* p<0.01.
†p<0.05.
‡p<0.1.
GYTS, Global Youth Tobacco Surveys; RIP, relative income price.

never-smokers), which produced a sample of 1611 adolescents with similar characteristics. An adolescent subjected to a unit increase in RIP has a 0.97 times lower chance of initiating smoking (OR=0.974, p<0.01) compared with those not exposed to the same increase in RIP. Similarly, in the matched sample of the GYTS 2017 (table 3), a unit increase in RIP is associated with 0.80 times lower odds of smoking initiation (OR=0.804, p<0.01).

Getting free cigarettes (OR=0.517, p<0.01), family/class discussion on tobacco (OR=1.711, p<0.01) and hearing antismoking messages (OR=2.048, p<0.01) are all found to be statistically significant in influencing smoking initiation in the matched sample (table 2). However, the odds for these variables are contrary to *a priori* expectations. Similarly, in the matched sample (table 2), the likelihood of initiating smoking among women is lower. The results imply that women have about 39% lower probability of initiating smoking (OR=0.615, p<0.01) than men.

### Affordability elasticities

In the unmatched sample, the estimated affordability elasticity is −0.372 (CI −0.701 to −0.042) for the 2000–2009 sample and −1.247 (−2.248 to −0.246) for the 2017 sample. These elasticities are statistically significant at the 5% level. By sex, the affordability elasticity is −0.137 for men and −0.888 for women for the 2000–2009 sample, but only that of women is statistically significant (p<0.01).

**Table 4** Affordability elasticity estimates

| | Both sexes | | Men | | Women | |
|---|---|---|---|---|---|---|
| | Unmatched | Matched | Unmatched | Matched | Unmatched | Matched |
| **Panel A: 2000–2009** | | | | | | |
| Variables | Unmatched | Matched | Unmatched | Matched | Unmatched | Matched |
| Affordability | −0.372† | −0.490* | −0.137 | −0.326 | −0.888* | −0.928* |
| | (0.168) | (0.168) | (0.219) | (0.216) | (0.253) | (0.258) |
| 95% CI | −0.701 to −0.042 | −0.818 to −0.161 | −0.567 to 0.292 | −0.749 to 0.097 | −1.384 to −0.392 | −1.434 to −0.422 |
| Observations | 106 673 | 10 078 | 55 396 | 5648 | 51 277 | 4430 |
| **Panel B: 2017 GYTS** | | | | | | |
| Affordability | −1.247† | −1.349* | −0.938† | −1.045† | −1.610‡ | 1.518† |
| | (0.511) | (0.500) | (0.474) | (0.484) | (0.866) | (0.778) |
| 95% CI | −2.248 to −0.246 | −2.328 to −0.369 | −1.867 to −0.008 | −1.993 to −0.096 | −3.307 to −0.087 | −3.043 to −0.007 |
| Observations | 37 654 | 4850 | 18 084 | 2807 | 19 570 | 2043 |

Standard errors in parentheses.
*p<0.01.
† p<0.05.
‡ p<0.1.
GYTS, Global Youth Tobacco Surveys.

The elasticities are higher for 2017 GYTS (−0.938 for men and −1.610 for women). The results are presented in table 4.

In the matched sample for the 2000–2009 GYTS, the overall elasticity is −0.490 (CI −0.818 to −0.161) for both sexes (table 4), which is similar to that of the unmatched sample. For men, the effect of changes in RIP is statistically insignificant. Among women, a percentage increase in RIP is associated with a 0.928% lower probability of smoking initiation. The elasticities for both men and women in the 2017 GYTS were negative, statistically significant, and more than unity.

The mean and median standardised difference for the covariates used in matching show that the matching satisfies the balancing test (results not reported). The mean and median standardised difference is 2.3%, which is below the normal 10% threshold. Therefore, the balancing property is satisfied.

## DISCUSSION AND CONCLUSION

In this study, increasing the RIP of cigarettes is significantly associated with a lower probability of initiating smoking. This finding is consistent with international literature, including the few existing studies in SSA that have reported that making cigarettes less affordable lowers the likelihood of smoking initiation among young people.[17 18 26] In addition, the results from the unmatched 2000–2009 sample suggest that men are not responsive to changes in RIP while women are. Nevertheless, in the matched sample analysis, especially using the 2017 GYTS, both men and women are responsive to changes in RIP. Indeed, the issue of affordability becomes more important given that Ghana's per capita income has been growing at an average of 4.4% annually in the last decade.[30]

Parental smoking increased the odds of smoking initiating. This points to the parental influence on the lifestyle of adolescents. Adolescents whose parents smoke may perceive smoking as acceptable behaviour. Previous studies have reported similar findings.[10 27 31 53] The odds of smoking initiation are higher for those whose friends' smoke. This points to the influence of peers, and is consistent, for example, with Mak et al.[31] Those who were offered free cigarettes by sales representatives were more likely to initiate smoking. All forms of tobacco promotion and advertising were banned in Ghana in 1982. However, the tobacco industry seems to be breaking these laws, because 12.44% of youth reported being offered a cigarette for free. This observation, together with our results, suggests the need to strengthen the enforcement of the existing ban on all forms of tobacco advertising and promotion in Ghana.

This study has several limitations. The GYTS is a self-reporting survey, which means that the responses are prone to recall bias even in cases where the adolescents are required to answer questions about events that occurred in the past 30 days. For instance, students may not recall the exact age at which they tried their first puff. There is also a social desirability bias when self-reporting behaviours such as smoking, especially among women. The weakness of our measure of smoking initiation is that it may not predict regular smoking behaviour well.[42 43] In addition, there are other important factors affecting smoking uptake among the youth that are not included in this study. For instance, changing community norms regarding smoking, the enforcement of laws regarding the sale of cigarettes to minors and changes in the social image of smoking are key factors that may influence smoking participation,[54 55] but are not included in the models estimated.

In conclusion, making cigarettes less affordable is associated with a lower probability of smoking initiation among adolescents in Ghana. This supports the use of

price measures, through higher excise taxes, as effective strategies to decrease smoking initiation among adolescents in Ghana. Since incomes are rising at the average of 4.4% annually,[30] tobacco taxes need to be adjusted regularly to ensure that cigarettes or other tobacco products become less affordable over time in order to discourage young people from initiating smoking and to encourage smokers to quit.

**Author affiliations**
[1]SAMRC/Wits Centre for Health Economics and Decision Science — PRICELESS SA, School of Public Health, University of the Witwatersrand, Johannesburg, Gauteng, South Africa
[2]Institute of Health Research, University of Health and Allied Sciences, Ho, Volta Region, Ghana
[3]Department of Economics, University of Namibia, Windhoek, Namibia
[4]Department of Health Sciences, University of York, York, UK
[5]Research Unit on the Economics of Excisable Products (REEP), School of Economics, University of Cape Town, Rondebosch, Western Cape, South Africa

**Contributors** Conceptualisation: MKB; methodology: MKB; software: MKB; validation: MKB, MI, ENT, NDM and HR; formal analysis: MKB; investigation: MKB, MI, ENT, NDM and HR; data curation: MKB; writing—original draft preparation: MKB, MI, ENT, NDM and HR; writing—review and editing: MKB, MI, ENT, NDM, HR; project administration: MKB; supervision: HR; funding acquisition: HR, NDM; guarantor: MKB. All authors have read and approved the manuscript.

**Funding** Funding for this study was provided by Cancer Research UK (Grant number: C62640/A24723) and International Development Research Centre (Grant Number: 108820 - 001) with additional support from the SAMRC/Wits Centre for Health Economics and Decision Science – PRICELESS SA (Grant Number 23108). Funding for ENT and NDM is acknowledged from the Tobacco Control Capacity Programme (Grant Number: MR/P027946/2) supported by UK Research and Innovation (UKRI) with funding from the Global Challenges Research Fund (GCRF). The funders had no role in study design, data collection and analysis, or preparation of the manuscript.

**Competing interests** None declared.

**Patient and public involvement** Patients and/or the public were not involved in the design, or conduct, or reporting, or dissemination plans of this research.

**Patient consent for publication** Not applicable.

**Ethics approval** This study does not involve human participants.

**Provenance and peer review** Not commissioned; externally peer reviewed.

**Data availability statement** Data are available in a public, open access repository. Data are available upon reasonable request. The publicly available data can be accessed: https://nccd.cdc.gov/GTSSDataSurveyResources/Ancillary/DataReports.aspx?CAID=2. The relative income price data are available from the authors on reasonable request.

**ORCID iDs**
Micheal Kofi Boachie http://orcid.org/0000-0003-1062-889X
Hana Ross http://orcid.org/0000-0001-5799-1915

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
