## [Reviewer comments · BMJ Open]

ARTICLE DETAILS

TITLE (PROVISIONAL)	Effect of relative income price on smoking initiation among adolescents in Ghana: evidence from pseudo-longitudinal data
AUTHORS	Boachie, Micheal Kofi; Immurana, Mustapha; Tingum, Ernest; Mdege, Noreen; Ross, Hana

VERSION 1 – REVIEW

REVIEWER	Baum, Christopher Boston College, Economics
REVIEW RETURNED	16-Sep-2021

GENERAL COMMENTS	Review of “The association between relative income price and smoking initiation among adolescents in Ghana” The authors use several waves of a survey of adolescent behavior to investigate the likelihood of smoking initiation. They employ a discrete-time hazard model to evaluate the impact of a number of factors on initiation in a logistic regression context, as well as forming a propensity-score-matched sample of smokers and nonsmokers. Their empirical strategy appears sound, and the conclusions drawn from their modeling in the contexts of elasticities are reasonable. I am surprised that both models fail to find that age of the individual is related to initiation. The major concern I have with the paper is its reliance on survey data from 2000, 2006 and 2009 rather than any more recent study of adolescent behavior. Is it the case that the GYTS survey was not conducted in more recent years? They acknowledge that the 2017 data are not comparable due to ‘inconsistencies in the questions asked’ but do not indicate whether there were any other waves of the survey available. As there have been tobacco control initiatives worldwide in the last 10–15 years, any conclusions drawn from this study through 2009 miss whatever impact those initiatives may have had in Ghana. At a very least, I believe the study should be replicated to the extent possible on the 2017 survey. As it stands, the paper is of
--

	some historical interest, but fails to add to the literature that has evaluated anti-smoking measures in the past decade. One glaring typo: the RIP is defined as the percentage of GDP required to buy 100 packs of 2000 cigarettes. As cigarettes are usually marketed in packs of size 20, I presume the authors mean the cost of a total of 2000 cigarettes.
--	---

REVIEWER	Ma, Shaoying OSUMC, Center for Tobacco Research
REVIEW RETURNED	15-Nov-2021

GENERAL COMMENTS	The authors examine the effect of cigarette affordability on smoking initiation among adolescents in Ghana. The affordability measure that the authors use is relative income price (abbreviated as RIP), and this study employs data from the Global Youth Tobacco Surveys (GYTS). The authors find that the likelihoods of adolescent smoking initiation is inversely related to RIP in Ghana, and they report elasticities for the whole sample as well as for men and women separately. This study implies that as income grows, taxation might be an effective policy tool to curb cigarette use in Ghana. This paper contributes to the literature by being one of the few studies that present evidence of the effect of cigarette affordability on smoking initiation in Sub-Saharan Africa. The paper is well-written and I enjoyed reading it. Please see below for specific comments.  1. Page 6 lines 53-55, does Ghana have minimum age law on cigarettes? Might be helpful to provide the policy context. 2. Page 7 lines 29-34, please add citations for the changes in taxation. 3. Page 7 line 46, how do the authors address the endogeneity? When reading this, I thought the authors are going to estimate causal relationship; but then this sentence was followed by "examine... association...", which is consistent with what I read from the abstract. I just think it's a bit confusing here, and the authors should talk about the empirical approaches they take in the introduction section. 4. Page 8 line 10, although the authors describe the data they use in the methods section, I think it would be helpful to also briefly mention that information in the introduction section, such as "We hypothesize that ..., and make use of the Global Youth Tobacco Survey (GYTS) data." 5. Page 8 line 31, I'm still confused by the age range of respondents in GYTS data that the authors use; is it 11-17 years old? Or does GYTS questionnaire classify respondents into three groups, ≤ 11, 11-17, ≥ 17, and the authors only include the middle group (11-17)? 6. Page 9 line 34, it might be better to say "100 packs of cigarettes (20 sticks per pack, in total 2,000 sticks)". 7. Page 9 line 46, please add citation(s) for "publications of the tobacco industry and the Government of Ghana". 8. Page 9 line 57, the authors indicate that they use three waves of GYTS data (2000, 2006, 2009) and RIP data from 1991-2009, at the beginning of the methods section. For the study period outside of 2008-2018, how do the authors obtain RIP data? It would be helpful to elaborate a bit here.
--

	9. Page 10 line 21, if I'm understanding correctly, the GYTS data that the authors use are pooled cross-sectional from three years. Please go into a bit more details to explain how the pseudo longitudinal sample is constructed, size of subsample that the authors are able to follow over time, etc. 10. Page 12 line 12, the prevalence of smoking based on the surveys seems to be much higher than the statistics on page 6 line 18. Why is that? Is it because the authors use ever smoking to calculate smoking prevalence in the survey and the statistics on page 6 line 18 use a different definition? 11. Page 13 Table 2, I wonder does GYTS ask geographic locations of respondents and if there is regional variation in tobacco policies in Ghana? If both are "yes", the authors might want to cluster standard errors at the regional level. In addition, the authors should add year fixed effects. 12. Page 17, discussion section, overall I find the discussion section very well written and quite interesting and informative. The authors could cite some statistics of GDP growth and time trend of smoking prevalence in Ghana, to provide more context for the readers and strengthen their argument in the last paragraph of the discussion section.
--	---

VERSION 1 – AUTHOR RESPONSE

Reviewer 1

Review of "The association between relative income price and smoking initiation among adolescents in Ghana"

The authors use several waves of a survey of adolescent behavior to investigate the likelihood of smoking initiation. They employ a discrete-time hazard model to evaluate the impact of a number of factors on initiation in a logistic regression context, as well as forming a propensity-score-matched sample of smokers and nonsmokers. Their empirical strategy appears sound, and the conclusions drawn from their modeling in the contexts of elasticities are reasonable. I am surprised that both models fail to find that age of the individual is related to initiation.

Comment 1

The major concern I have with the paper is its reliance on survey data from 2000, 2006 and 2009 rather than any more recent study of adolescent behavior. Is it the case that the GYTS survey was not conducted in more recent years? They acknowledge that the 2017 data are not comparable due to 'inconsistencies in the questions asked' but do not indicate whether there were any other waves of the survey available. As there have been tobacco control initiatives worldwide in the last 10-15 years, any conclusions drawn from this study through 2009 miss whatever impact those initiatives may have had in Ghana. At a very least, I believe the study should be replicated to the extent possible on the 2017 survey. As it stands, the paper is of some historical interest, but fails to add to the literature that has evaluated anti-smoking measures in the past decade.

Author response 1

Thank you for this observation. In fact, we have been searching for a more comparable data on adolescents in Ghana. However, we found no other survey on adolescents with comparable measures except the ones outlined. We agree that new tobacco control policies have been implemented in recent years (such as the ban on sale to children). However, the results from previous surveys allows us to better understand how affordability impacts on smoking behavior going forward. As suggested by Dr Baum, we have analysed the 2017 survey separately and we find the results on RIP or affordability to be similar to the previous data.

Comment 2

One glaring typo: the RIP is defined as the percentage of GDP required to buy 100 packs of 2000 cigarettes. As cigarettes are usually marketed in packs of size 20, I presume the authors mean the cost of a total of 2000 cigarettes.

Author response 2

Thank you for this observation. Yes, we are referring to the cost of 2000 cigarettes. We have revised the sentence, based on your observation and that of the other reviewer, as: The main independent variable is the affordability index or the RIP, measured as the percentage of GDP per capita (per capita income) required to buy 100 packs of cigarettes (20 sticks per pack, in total 2,000 sticks).

Reviewer: 2

Dr. Shaoying Ma, OSUMC

Comments to the Author:

The authors examine the effect of cigarette affordability on smoking initiation among adolescents in Ghana. The affordability measure that the authors use is relative income price (abbreviated as RIP), and this study employs data from the Global Youth Tobacco Surveys (GYTS).

The authors find that the likelihoods of adolescent smoking initiation is inversely related to RIP

in Ghana, and they report elasticities for the whole sample as well as for men and women separately. This study implies that as income grows, taxation might be an effective policy tool to curb cigarette use in Ghana. This paper contributes to the literature by being one of the few studies that present evidence of the effect of cigarette affordability on smoking initiation in Sub-Saharan Africa. The paper is well-written and I enjoyed reading it.

Please see below for specific comments.

Comment 1

1. Page 6 lines 53-55, does Ghana have minimum age law on cigarettes? Might be helpful to provide the policy context.

Author response 1

Thank you for this comment. Prior to 2012 when the Public Health Act was introduced there was no minimum age law. However, the new Act restricts the sale of tobacco to persons below age 18. We have included this in the manuscript (see page 8).

Comment 2

2. Page 7 lines 29-34, please add citations for the changes in taxation.

Author response 2

We have included the citation for the tax changes (see page 6).

Comment 3

3. Page 7 line 46, how do the authors address the endogeneity? When reading this, I thought the authors are going to estimate causal relationship; but then this sentence was followed by “examine... association...”, which is consistent with what I read from the abstract. I just think it’s a bit confusing here, and the authors should talk about the empirical approaches they take in the introduction section.

Author response 3

We have added the empirical approach taken in the last paragraph of the introductory section (see page 7).

Comment 4

4. Page 8 line 10, although the authors describe the data they use in the methods section, I think it would be helpful to also briefly mention that information in the introduction section, such as “We hypothesize that ..., and make use of the Global Youth Tobacco Survey (GYTS) data.”

Author response 4

We have revised the section to include “We hypothesize that making cigarettes less affordable reduces the likelihood of smoking initiation among young people, and make use of the Global Youth Tobacco Survey (GYTS) data and other datasets to test that hypothesis” (see page 6).

Comment 5

5. Page 8 line 31, I’m still confused by the age range of respondents in GYTS data that the authors use; is it 11-17 years old? Or does GYTS questionnaire classify respondents into three groups, ≤ 11 , 11-17, ≥ 17 , and the authors only include the middle group (11-17)?

Author response 5

The GYTS questionnaire classifies respondents into 11 years or younger; 12, 13, 14, 15, 16 and 17 years or older. This is the age of the respondents as at the time of the survey.

The above age classification is different from when the respondent first smoked. The age that the respondent first smoked starts at 7 years or younger and ends at 16 years or older. We use both age at the time of survey and age at first smoke to determine which year the person smoked. Therefore, we start the follow up when children are age 8 until they smoke and exit the sample, or till the time the study ends. So the age range is 11 years or younger to 17 years or higher at the time of the survey.

Comment 6

6. Page 9 line 34, it might be better to say “100 packs of cigarettes (20 sticks per pack, in total 2,000 sticks)”.

Author response 6

We have revised this based on your suggestion (see page 9).

Comment 7

7. Page 9 line 46, please add citation(s) for “publications of the tobacco industry and the Government of Ghana”.

Author response 7

We have provided the citations (see page 9).

Comment 8

8. Page 9 line 57, the authors indicate that they use three waves of GYTS data (2000, 2006, 2009) and RIP data from 1991-2009, at the beginning of the methods section. For the study period outside of 2008-2018, how do the authors obtain RIP data? It would be helpful to elaborate a bit here.

Author response 8

The price data from 2008 and 2009 come directly from WHO and Government of Ghana's report card on the FCTC in 2009. Based on the price point for 2008 and tobacco specific inflation we performed a backward calculation of the price for 2007 as follows:

$$P_{t-1} = \frac{P_t}{(1 + Tob.Inflation_t)}$$

The year 2000 and 2006 retail prices are also calculated following that procedure. This approach is similar to the approach used by Asare et al. (2019). We then calculated the RIP and used that for analysis. So for the period outside 2008 - 2018, we use the above formula to obtain the price and later RIP (see page 9).

Comment 9

9. Page 10 line 21, if I'm understanding correctly, the GYTS data that the authors use are pooled cross-sectional from three years. Please go into a bit more details to explain how the pseudo longitudinal sample is constructed, size of subsample that the authors are able to follow over time, etc.

Author response 9

The difference between current age and age at first smoke gives the number of years the person has smoked. STATA routine command, `expand`, is used to expand the person's age at the time of the survey after which an event variable indicating smoking status is created. Students who are below age 8 and who started smoking before age 8 are removed from the dataset (see page 10). By using the `expand` command, one is able to obtain the year of birth.

Comment 10

10. Page 12 line 12, the prevalence of smoking based on the surveys seems to be much higher

than the statistics on page 6 line 18. Why is that? Is it because the authors use ever smoking to calculate smoking prevalence in the survey and the statistics on page 6 line 18 use a different definition?

Author response 10

The prevalence statistics on Page 12 uses ever smoking for all the three waves of the data and then counts the number of those who have ever smoked. The other statistics on page 6 is the very recent statistics, which does not make use of a pooled data from previous waves. So yes, differences in definition is the reason for the differences seen.

Comment 11

11. Page 13 Table 2, I wonder does GYTS ask geographic locations of respondents and if there is regional variation in tobacco policies in Ghana? If both are “yes”, the authors might want to cluster standard errors at the regional level. In addition, the authors should add year fixed effects.

Author Response 11

The publicly available GYTS dataset reports no geographic locations and therefore we are unable to account for regional variations. Because Ghana uses a unitary governance structure, there is no regional variation in tobacco laws as all laws apply to all parts of the country. Concerning year fixed effects, we included time variable and indicator variable for survey year as suggested by Professor Jenkins and other previous studies.^{1 2} We believe that the methodology has adequately captured the effect of time.

Comment 12

12. Page 17, discussion section, overall, I find the discussion section very well written and quite interesting and informative. The authors could cite some statistics of GDP growth and time trend of smoking prevalence in Ghana, to provide more context for the readers and strengthen their argument in the last paragraph of the discussion section

Author response 12

We have included some GDP statistics in the introduction section and also in the discussion.

VERSION 2 – REVIEW

REVIEWER	Ma, Shaoying OSUMC, Center for Tobacco Research
REVIEW RETURNED	20-Dec-2021

GENERAL COMMENTS	This is a revised version of the paper that I reviewed in November, and the authors study the impact of cigarette affordability (measured by relative income price, i.e. RIP) on youth initiation in Ghana, using data from Global Youth Tobacco Surveys (GYTS). The authors made some major changes to the original version of their manuscript, which much improved the paper. Please see below for specific comments. 1. Page 9 line 13, I appreciate that the authors added more details about how they addressed the endogeneity issue, and I wonder how does “iii) the fact that some people will never choose to smoke or use any form of tobacco, for example, for reasons of health or religious belief” help to alleviate the endogeneity concern?2. Page 10 line 40, “amount spent on tobacco in the last 30 days before the survey” Is it money amount, time amount, or something else?3. Page 10 line 12 and page 11 line 60, “age-at-risk criteria” I see that “age-at-risk criteria” was explained in more details on page 12 lines 19-26. I would recommend moving the explanation in the text, and explain it when it is first mentioned, i.e. page 10 line 12; otherwise readers may not understand the basis of choosing to start “from age 8 (age-at-risk criteria)”. Also citations should be added to “age-at-risk criteria” on page 10 line 12 and page 11 line 60.
---

VERSION 2 – AUTHOR RESPONSE

Reviewer: 2

Dr. Shaoying Ma, OSUMC

Comments to the Author:

This is a revised version of the paper that I reviewed in November, and the authors study the impact of cigarette affordability (measured by relative income price, i.e. RIP) on youth initiation in Ghana, using data from Global Youth Tobacco Surveys (GYTS).

The authors made some major changes to the original version of their manuscript, which much improved the paper.

Author response

Thank you very much for the suggestions.

Please see below for specific comments.

Comment

1. Page 9 line 13, I appreciate that the authors added more details about how they addressed the endogeneity issue, and I wonder how does “iii) the fact that some people will never choose to smoke or use any form of tobacco, for example, for reasons of health or religious belief” help to alleviate the endogeneity concern?

Author response

We have deleted the third item “iii) the fact that some people will never choose to smoke or use any form of tobacco, for example, for reasons of health or religious belief” Now the endogeneity is tackled using the aggregate prices and propensity score matching.

Comment

2. Page 10 line 40, “amount spent on tobacco in the last 30 days before the survey”

Is it money amount, time amount, or something else?

Author response

Thank you very much for the observation. We were referring to money spent on tobacco. Thus, the expenditure on tobacco. We have revised it as “... money spent on tobacco ... “

Comment

3. Page 10 line 12 and page 11 line 60, “age-at-risk criteria”

I see that “age-at-risk criteria” was explained in more details on page 12 lines 19-26. I would recommend moving the explanation in the text, and explain it when it is first mentioned, i.e. page 10 line 12; otherwise readers may not understand the basis of choosing to start “from age 8 (age-at-risk criteria)”. Also citations should be added to “age-at-risk criteria” on page 10 line 12 and page 11 line 60.

Author response

Thank you for the suggestion. We have revised the section and also included the citations at the first mention of the age-at-risk criteria. The text now reads as (page 8):

Although the GYTS data contain adolescents whose first puff was at age 7 or younger, we assume that a student is at risk at age 8 because that is the age at which the child is relatively developed and is able to start out-of-home interaction with peers.[1-3] Students who started smoking before reaching age 8 and those below age 8 are therefore excluded from the pooled sample and not followed. In line with previous studies, a student exits the sample once smoking is initiated.[1,2,4,5]